# From Trypomastigotes to Trypomastigotes: Analyzing the One-Way Intracellular Journey of *Trypanosoma cruzi* by Ultrastructure Expansion Microscopy

**DOI:** 10.3390/pathogens13100866

**Published:** 2024-10-02

**Authors:** Ramiro Tomasina, Fabiana C. González, Andrés Cabrera, Yester Basmadjián, Carlos Robello

**Affiliations:** 1Laboratorio de Interacciones Hospedero Patógeno, Institut Pasteur de Montevideo, Montevideo 11400, Uruguay; rtomasina@pasteur.edu.uy (R.T.); fcgonzalez@pasteur.edu.uy (F.C.G.); cabrera@pasteur.edu.uy (A.C.); 2Unidad Académica de Parasitología y Micología, Facultad de Medicina, Universidad de la República, Montevideo 11550, Uruguay; yester@higiene.edu.uy; 3Unidad Académica de Bioquímica, Facultad de Medicina, Universidad de la República, Montevideo 11550, Uruguay

**Keywords:** *Trypanosoma cruzi*, UExM, cell division, amastigogenesis, trypomastigogenesis

## Abstract

The protozoan parasite *Trypanosoma cruzi* is the causative agent of Chagas disease, also called American trypanosomiasis. This neglected tropical disease affects millions of individuals across the Americas. To complete its life cycle, *T. cruzi* parasitizes both vertebrate hosts and its vector, commonly known as the ‘kissing bug’. The parasite’s survival and proliferation strategies are driven by the diverse environments it encounters. Despite being described by Carlos Chagas in 1909, significant knowledge gaps persist regarding the parasite’s various life forms and adaptive capabilities in response to environmental cues. In this study, we employed Ultrastructure Expansion Microscopy to explore the intricate journey of *T. cruzi* within the host cell. Upon entry into the host cell, trypomastigotes undergo folding, resulting in intermediate forms characterized by a rounded cell body, anterior positioning of basal bodies, and a shortened flagellum. The repositioning of basal bodies and the kinetoplast and the shortening of the flagella mark the culmination of intracellular amastigogenesis. Furthermore, we analyzed intracellular trypomastigogenesis, identifying discrete intermediate forms, including leaf-shaped stages and epimastigote-like forms, which suggests a complex differentiation process. Notably, we did not observe any dividing intracellular epimastigotes, indicating that these may be non-replicative forms within the host cell. Our detailed examination of amastigote cell division revealed semi-closed nuclear mitosis, with mitotic spindle formation independent of basal bodies. This study provides new insights into the morphological and cytoskeletal changes during the intracellular stages of *T. cruzi*, providing a model for understanding the dynamics of intracellular amastigogenesis and trypomastigogenesis.

## 1. Introduction

The discovery of *Trypanosoma cruzi* by Carlos Chagas and the description of the disease, its life cycle, and its hosts [1] constitute a milestone in parasitology and protozoology. Paradoxically, one of the advantages of Chagas’s work was the absence of powerful tools for microscopic analysis, which led to the fact that an important part of this work is based on a deep observation of the morphology of *Trypanosoma cruzi* and its different forms. Although the nomenclature proposed by Chagas was not subsequently adopted—mainly because the work exhibits a bias towards malaria—the morphological variability of *T. cruzi*, depending on the origin of the sample, is remarkable. Similarly, work on other trypanosomatids showed comparable results: the capacity for morphological changes depending on different environments is characteristic of trypanosomatids. This is well summarized by Vickerman et al. [2], who reported that trypanosomatids can present at least the following forms: promastigotes, amastigotes, trypomastigotes, epimastigotes, spheromastigotes, opistomastigotes, and choanomastigotes. The advent of more powerful imaging tools allowed for a more detailed analysis of the *T. cruzi* cell cycle, its organelles, and cellular organization [3,4,5,6,7,8,9].

Within the vertebrate host, *T. cruzi* exhibits two primary life stages: non-replicative trypomastigotes characterized by an elongated flagellum, and replicative amastigotes [3], which have a rounded morphology and a shortened flagellum and undergo replication until they revert to the trypomastigote stage, thereby completing the lytic cycle [3,10]. Vector infection occurs when a triatomine insect ingests blood containing trypomastigotes from an infected vertebrate. Within the vector, the parasite transitions to the epimastigote form, a flagellated and replicating stage residing in the insect’s intestine [3,10]. Subsequently, the epimastigote matures into the metacyclic trypomastigote, a non-replicative and infective stage in the bug’s feces [3]. Transmission occurs when infected vectors deposit feces onto the skin or mucosa of a vertebrate during a blood meal, facilitating infection and completing the cycle [10]. However, the original observations by Carlos Chagas over a century ago hinted at a more intricate life cycle than four life forms. Remarkable work using transmission electron microscopy (TEM) and scanning electron microscopy (SEM) has been conducted over the years [4,8,9,11,12], shedding light on these different life forms. In terms of this, high-resolution microscopy studies have revealed that in the transition of epimastigotes into metacyclic trypomastigotes, three intermediate stages exist preceding the elongated metacyclic trypomastigote shape [4].

Studies on the intracellular life forms within vertebrate hosts have unveiled intermediate stages that share similarities with epimastigotes [12,13,14]. These intracellular epimastigotes exhibit the same positioning of the basal bodies and kinetoplast relative to the nucleus and possess a body shape similar to their extracellular counterparts [13,14]. However, the intracellular epimastigotes are five times smaller and can infect cells [14]. Recent advances have revealed metabolically slower, persistent entities termed ‘dormant life stages’ that withstand drug treatments [15]. The triggers prompting parasites to adopt a dormant state or progress through the life cycle to trypomastigotes remain elusive. Nonetheless, dormant parasites can revert to an active life cycle and reach infective stages [15].

In the vertebrate host, upon invading the cells, trypomastigotes, encapsulated by a parasitophorous vacuole [3,16,17,18,19], initiate their transformation into the rounded, replicative form known as the amastigote. However, our understanding of intracellular amastigogenesis and the reverse process, trypomastigogenesis, remains limited due to the scarcity of high-resolution imaging data available for these intracellular differentiation processes. These mechanisms imply trypomastigote transitions from a long-flagellated stage to a rounded replicating form with a short flagellum and back to a non-replicative form with a long flagellum, still representing a significant knowledge gap in *T. cruzi* biology. In this work, we employ Ultrastructure Expansion Microscopy (UExM) to elucidate the complex journey of *T. cruzi* along the host.

## 2. Materials and Methods

### 2.1. Cell Culture

Vero cells were maintained in Dulbecco’s Modified Eagle Medium (DMEM GibcoTM, Thermo Fisher Scientific, Waltham, MA, USA), supplemented with 10% heat-inactivated bovine serum (GibcoTM, Thermo Fisher Scientific, Waltham, MA, USA) and penicillin/streptomycin (Invitrogen, Carlsbad, CA, USA) at 37 °C in a 5% CO_2_ atmosphere. For infection assays, trypomastigotes of the Dm28c strain of *T. cruzi* were incubated with semiconfluent Vero cells (5:1 parasite/cell ratio) for one hour.

### 2.2. Ultrastructure Expansion Microscopy

Ultrastructure Expansion Microscopy (UExM) was implemented adhering to the established methodology for *T. cruzi* and other protozoans, as delineated previously [20,21,22,23,24], ensuring fidelity to the original protocol. Briefly, coverslips containing cells hosting parasitic infections underwent a 5 h incubation at 37 °C with a mixture comprising 0.7% acrylamide (AA) and 1% formaldehyde (FA). Subsequently, gelification was commenced by introducing a solution of 19% sodium acrylate (SA), 10% AA, and 0.1% BIS-AA in PBS, catalyzed over 1 h at 37 °C. Proteins were then denatured through a 1.5 h incubation at 95 °C, facilitating the expansion of the gel enclosing the parasites, which was subsequently immersed in water for overnight expansion. As previously reported, the expansion factor obtained was ~3.5–4.0 [20,21,22,23,24].

A conventional indirect immunofluorescence protocol was employed to visualize the parasites’ ultrastructure, as broadly used [20,21,22,23,25]. Mouse anti-acetylated tubulin (a marker for acetylated tubulin structure) (Sigma T7451) and goat anti-mouse Alexa Fluor 488 (Invitrogen) were both utilized at a dilution of 1:500 in PBS. NHS ester 594 (Invitrogen) and DAPI (Invitrogen) were used to label primary amines on proteins and the genetic material, respectively. NHS ester 594 was used at a dilution of 1:250 in PBS and DAPI at 1 μg/mL in PBS.

Imaging was conducted using a Zeiss LSM800 confocal microscope (Oberkochem, Germany) with a Plan-Apochromat 63×/1.40 oil immersion objective and a 560 axiocam mono camara. Post-image acquisition and processing were performed using ImageJ v1.54g (NIH) and Zeiss ZEN blue edition v2.0 software. The 3D model presented in SFig1 was constructed using Agave v1.5.0 freeware.

## 3. Results

### 3.1. Intracellular Amastigogenesis and Basal Body Repositioning

To study the intracellular amastigogenesis process in detail, we analyzed the early infection stages using Ultrastructure Expansion Microscopy (UExM). It is well established that trypomastigotes initially adhere to the host cell, commencing the invasion process [18]. Our experimental approach allowed us to visualize this process and distinguish penetrating parasites and those within the cell (Figure 1).

After entry into the cell, the trypomastigotes undergo a defined sequence of morphological changes. Initially, the trypomastigote is surrounded by a parasitophorous vacuole while it folds around itself, allowing contact between its posterior and anterior parts. As this process occurs, an intermediate form characterized by a rounded cell body and a shortened flagellum arises (Figure 1). This intermediate form precedes the amastigote stage. It exhibits anterior positioning of basal bodies and the kinetoplast relative to the nucleus, but a long flagellum distinguishes it from mature amastigotes (Figure 1).

Four hours post-infection, rounded forms with shortened flagella were discernible. The positioning of basal bodies and the kinetoplast relative to the nucleus indicates that these forms correspond to amastigotes (Figure 2). Based on our observations, we propose a model of intracellular amastigogenesis in which trypomastigotes fold upon themselves, bringing the posterior end closer to the anterior end. This movement facilitates the repositioning of the basal body and kinetoplast from the posterior region (characteristic of trypomastigotes) to the anterior region, indicative of the amastigote stage, resulting in the culmination of the intracellular amastigogenesis process (Figure 2). This model implies membrane fusion and loss, as well as a reorganization of the subpellicular microtubules during this process. Unfortunately, data on subpellicular microtubule organization in *T. cruzi* are scarce [5,26]. However, studies from the early 1970s show that trypomastigote subpellicular microtubules are organized such that the anterior and posterior regions have fewer microtubules than the central part of the parasite [8]. Amastigotes and trypomastigotes differ in the number of subpellicular microtubules, with amastigotes containing 120–140, while trypomastigotes have 40–55 in the posterior part, 120–90 in the center, and 70–40 in the anterior part [8,26].

Considering that subpellicular microtubules may vary in length [5], it is plausible that microtubules from the posterior and anterior parts intercalate as membranes fuse (Figure 1). A similar model has been proposed to explain what is observed in extracellular amastigogenesis [11]. According to the authors, the membrane fusion needed for this model to work is consistent with the presence of membrane vesicles in the region of the flagellar tip and flagellar pocket [11].

### 3.2. Discrete Intermediate Forms Precede Intracellular Trypomastigogenesis

Next, we analyzed the process of intracellular trypomastigogenesis, beginning with amastigotes and culminating in mature trypomastigotes. We examined various time points, as intermediate forms have been previously documented [3,11,12,13,14]. Additionally, since amastigotes constitute the replicative intracellular stage of *T. cruzi,* we specifically explored the dynamics of the basal bodies as an indicative measurement of the parasites’ cell cycling.

At 24 h post-infection, three distinct life stages were identified, all exhibiting characteristics of dividing forms (Figure 3). Two stages displayed rounded morphologies, while a third showed a ‘leaf’ shape. Our observations indicate a sequential progression from short-flagellum, rounded stages (amastigotes and intermediate form I) to leaf-shaped forms (intermediate form II) during intracellular trypomastigogenesis (Figure 3). Notably, a helicoidal arrangement of microtubules (indicated by yellow arrows in Figure 3) is evident at this stage. Based on their morphology and localization, we assume that this arrangement of microtubules belongs to the cytostome–cytopharynx cytoskeleton, previously observed by high-resolution microscopy techniques [6,7,27]. Besides the morphological differences, the three distinct life stages meet the characteristics of amastigotes and have historically been identified as such [3,8,12].

By 72 h post-infection, diverse life forms were evident within the host cell, defying conventional amastigote and trypomastigote classifications (Figure 4). Leaf-shaped amastigotes and intermediate forms between amastigotes and longer flagella stages were observed initially. This was followed by other intermediate stages with elongated flagella but retaining a leaf-shaped body reminiscent of epimastigotes (intermediate form ‘epimastigote-like’, Figure 4). This form was previously observed and named ‘transitional epimastigote’ [12,13,14]. Consistent with what has been previously described, these ‘transitional epimastigotes’ are shorter than extracellular epimastigotes observed from LIT cultures [14]. Again, the helicoidal microtubule arrangement from the cytostome–cytopharynx cytoskeleton is also presented in this intracellular intermediate form and in the ‘epimastigote-like’ (yellow arrows in Figure 4).

Subsequently, an intermediate form with a longer flagellum and rounded cell body emerged, accompanied by a shift in the positioning of basal bodies towards the posterior, indicative of a precursor stage preceding trypomastigote formation (Figure 4). This intermediate form remains similar to the ‘drop epimastigote form’ observed during metacyclogenesis in vitro [4,28]. Finally, we could observe trypomastigotes in multiple orientations inside the host cell (Figure 4).

### 3.3. Are the Intracellular Epimastigotes a Non-Dividing Form?

Before the emergence of mature trypomastigotes, epimastigote-like forms were observed (Figure 4). This affirmation is based on the shape of the cell body and the position of flagella and basal bodies relative to the nucleus [12,13,14]. These forms were initially described as transitional epimastigotes [14] in a hybrid strain, and their presence in Dm28c indicates that they are part of the cycle independently of the *T. cruzi* lineage. Previously, work focused on studying this intermediate form has concluded that this ‘epimastigote-like’ is around five times shorter than its extracellular counterparts [14], which we could also observe in our work. Another notable aspect of these intracellular epimastigotes is that they are infective forms [29]. The similarities between the epimastigotes and the ‘transitional epimastigotes’ are significant, particularly in the positioning of the basal bodies and kinetoplast relative to the nucleus and the shape of the cell body. However, epimastigotes are actively dividing forms, which does not seem compatible with their intracellular localization. We analyzed them in detail, looking for dividing forms inside the host cell, but we could not detect dividing intracellular epimastigotes. We did not observe any intracellular epimastigotes with replicated basal bodies (Appendix A), which led us to propose that they may constitute non-replicative epimastigotes (Figure 4). Consistent with what was previously described, the presence of this intermediate form is relatively scarce compared with the total number of parasites observed [14]. This could reduce the number of dividing forms to nearly zero in the population, making it difficult to observe. It is interesting to highlight that, apparently, no matter the host, whether the vector or the vertebrate host, *T. cruzi* somehow needs to generate epimastigotes before differentiating into trypomastigotes.

### 3.4. Semi-Closed Nuclear Mitosis in Amastigotes

While the results obtained with intracellular epimastigotes are consistent with the fact that only amastigotes replicate during this life cycle phase, they suggest that tracking cell division becomes difficult when large numbers of parasites are inside the cell. Given that amastigotes are the replicative form, we performed a detailed analysis of cell division in amastigotes at an early stage of infection when cells are not heavily infected. At 24 h post-infection, before the onset of division, a single kinetoplast, a pair of basal bodies, and a single nucleus can be observed (Figure 5). At the initiation of division, both the basal bodies and the kinetoplast duplicate concurrently with DNA replication and the assembly of the mitotic spindle (panel II in Figure 5). Notably, this image reveals semi-closed nuclear mitosis in *T. cruzi*, with microtubule nucleation of the mitotic spindle occurring independently of basal bodies (panel II in Figure 5). A close examination shows that the nucleation of the mitotic spindle appears within a DAPI-unstained zone, reminiscent of previous observations using transmission electron microscopy (TEM) in extracellular replicative epimastigotes [30].

## 4. Discussion

Our study provides an in-depth imaging portrayal of the intricate life cycle of *Trypanosoma cruzi* within the host cell, with a particular focus on the series of morphological changes it undergoes from trypomastigote to amastigote and back to trypomastigote, using UExM. Understanding these transitions is crucial for unraveling the pathogenesis of Chagas disease, mainly because the way this parasite invades, replicates, and lyses out the cell, releasing a critical number of trypomastigotes, is the main reason for the spread of the infection through the organism infected [3,10,31].

Upon entry into the host cell, *T. cruzi* trypomastigotes undergo remarkable changes, transforming into the replicative form known as the amastigote [3,11,12,16,17,18,19,32]. This process involves a cascade of molecular events, including relocalization of the basal body and kinetoplast, followed by a shortening of the flagella [3,11,12] and the release of molecules to remove of the parasitophorous vacuole that initially surrounded it [11]. Our observations delineate the sequential order of these events, emphasizing the highly regulated nature of *T. cruzi* invasion within the host cell.

The transition from trypomastigote to amastigote marks a critical phase in the *T. cruzi* life cycle, as it is the stage where the parasite becomes capable of replication, proliferating within the host cell and occupying the tissue where it will remain or utilize for proliferation [3]. Our study presents a valuable microscopy imaging record that offers a detailed view of amastigogenesis, achieving a resolution not previously attained with immunofluorescence techniques (Figure 2). Based on our observations, we propose a model in which the trypomastigote, on its path to becoming an amastigote, reduces the size of its body and undergoes a rolling motion, bringing its anterior and posterior parts into proximity, facilitating the repositioning of the basal body and kinetoplast from the posterior part (trypomastigote) to the anterior part (amastigotes) (Figure 2). This model implies membrane fusion and membrane loss, as well as a reorganization of the subpellicular microtubules during this process. Unfortunately, data on subpellicular microtubule organization in *T. cruzi* are scarce [8,26]. Trypomastigote subpellicular microtubules are organized such that the anterior and posterior regions have fewer microtubules than the central part of the parasite [8]. Amastigotes and trypomastigotes differ in the number of subpellicular microtubules, with amastigotes containing 120–140, while trypomastigotes have 40–55 in the posterior part, 120–90 in the center, and 70–40 in the anterior part [8,26]. Given that subpellicular microtubules can vary in length [5], it is plausible that an intercalation of microtubules from the posterior and anterior parts could occur as membranes fuse (Figure 1). Pioneering electron microscopy studies [12] provide evidence of the intimate contact between the anterior and posterior part of trypomastigotes inside the cell. The presence of membrane vesicles attached to the flagellar tip and the flagellar pocket supports the idea of membrane fusion and membrane loss proposed in our model [11]. Once this process is completed, a rounded intermediate shape holding a long flagellum arises (Figure 1F). What the parasite does next on its road to becoming an amastigote remains unclear. We hypothesize that this intermediate form gradually internalizes the flagella to reach the complete amastigote morphology, similar to what was proposed for extracellular amastigogenesis [11].

Asymmetric division has been proposed as a model for amastigogenesis in *T. cruzi* [33]. According to this model, once the parasite is inside the cell, it forms a biflagellated entity that divides into two: a cell holding a kinetoplast without a nucleus called ‘zoid’ and an amastigote [34]. However, these forms have never been documented in other works. In line with this, in our work, we did not observe either the biflagellated cell or the zoid. Moreover, our research provides valuable insights into the mechanisms underlying amastigote replication. Using UExM, we observed semi-closed nuclear mitosis in *T. cruzi,* with microtubule nucleation occurring independently of basal bodies, similar to what is observed in epimastigotes [31]. We could also document three different morphologies among the amastigotes. The main differences are related to the flagella length and the shape of the cell body. It is interesting to highlight that the three meet the definition of amastigotes and, by regular optical microscopy, are indistinguishable.

A long-standing question in the field is whether the short flagella observed in amastigotes (Figure 3) originate from de novo synthesis or pre-existing flagella. Alves and Bastin recently discussed several scenarios for the formation of short flagella in amastigotes [33]. They propose that the short flagellum might arise from an early-locked flagellum, an equilibrium of assembly and disassembly with a high turnover of flagellar proteins, or a limitation in the pool of tubulin and other flagellar components [33]. In our study, we could not find any clue concerning that, but it is important to highlight that in all amastigotes, we invariably observed the presence of flagella.

The replication of amastigotes within the host cell is a critical step in the pathogenesis of Chagas disease, contributing to the dissemination of the parasite throughout the host’s tissues. Our study, employing high-resolution microscopy techniques, captures the sequential events leading to amastigote replication, providing a detailed understanding of the cellular dynamics underlying *T. cruzi* proliferation within the host. In addition, our findings shed light on the reverse transformation of amastigotes back into trypomastigotes, completing the life cycle of *T. cruzi* within the host cell. This process likely involves reorganizing cellular structures and activating specific molecular pathways to facilitate the transition from replicative to infective forms.

Overall, we provide a comprehensive analysis of the *T. cruzi* life cycle within the host cell and the series of morphological transformations from trypomastigote to amastigote and back to trypomastigote, as depicted in Figure 6 and Figure 7. By elucidating these transitions in detail, we advance our understanding of the cellular dynamics underlying *T. cruzi* replication and pathogenesis, laying the groundwork for future studies on this intriguing parasite.

## Figures and Tables

**Figure 1 pathogens-13-00866-f001:**
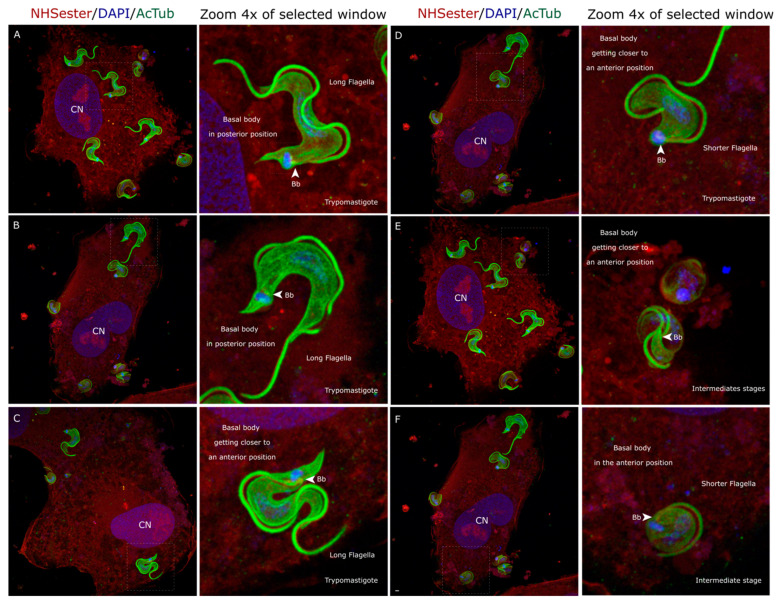
Intracellular trypomastigotes and rounded forms 30 min post-infection. (**A**–**C**) Trypomastigotes observed under UExM inside the cell. (**C**) Intermediate rounded shape form observed by UExM inside the cell. (**D**,**E**) In this rolling process, the parasite’s posterior part comes closer to its anterior part, making possible the position of the basal body and kinetoplast observed in the amastigotes. (**F**) In this panel, an intermediate rounded-shape life form is observed. Although similar to an amastigote form, it holds a longer flagellum. ‘Bb’ stands for basal bodies; ‘CN’ stands for cell nucleus. All the images are maximum-intensity projection z-stacks. The scale bar is 1 µm in all images.

**Figure 2 pathogens-13-00866-f002:**
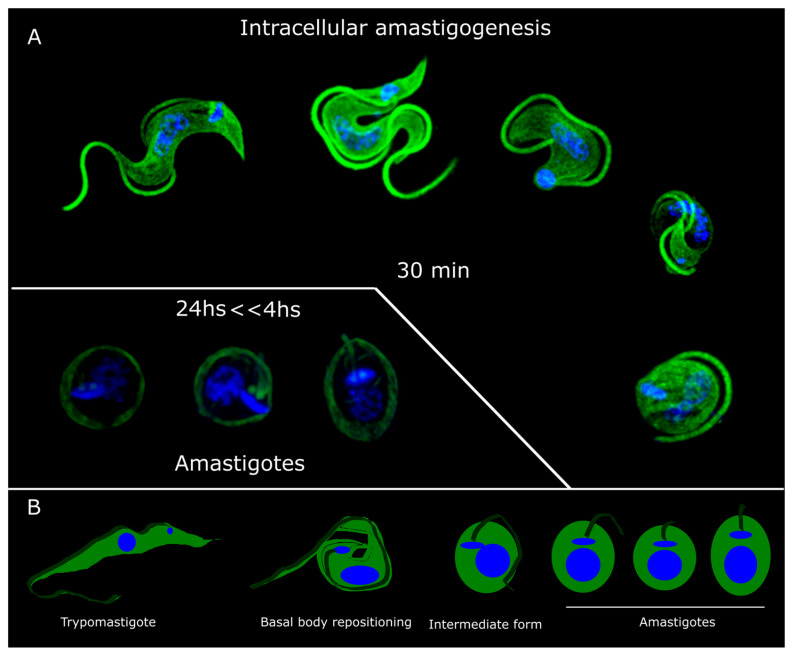
Intracellular amastigogenesis. (**A**) UExM of selected different life forms observed at 30 min post-infection and 4 and 24 h post-infection. All these life forms were observed in the process of the observation of amastigotes. (**B**) Amastigogenesis model. Upon entering the cell, the trypomastigotes undergo several transformations. These include shortening of the flagella and repositioning of the basal body and kinetoplast towards the nucleus. We propose a mechanism where the parasite executes a rolling movement, reduces its cell body size, and brings its posterior part in contact with the anterior part. Through a process of membrane fusion, *T. cruzi* forms a single enlarged rounded cell body, which subsequently gives rise to the amastigotes.

**Figure 3 pathogens-13-00866-f003:**
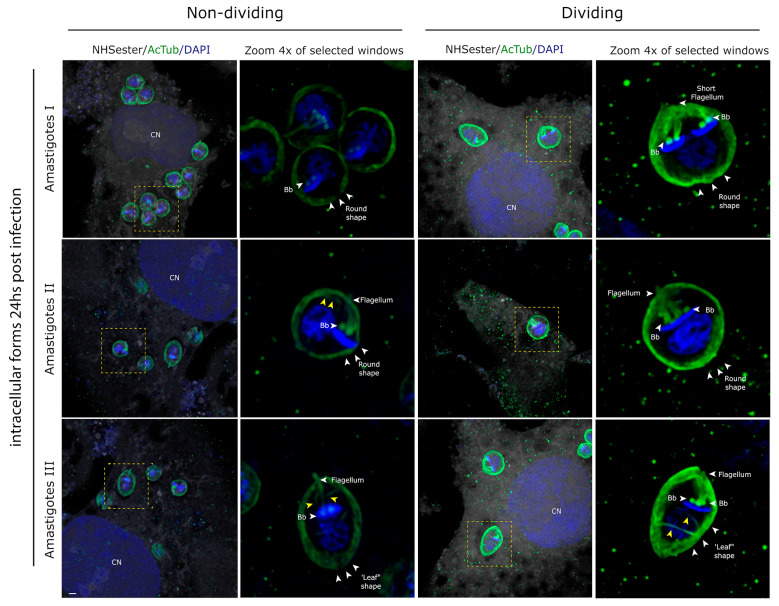
Three dividing forms are observed at 24 h post-infection. Using UExM, we could observe in detail the intracellular life form present at 24 h post-infection. Interestingly, although we assumed all of them were amastigotes, significant structural differences were observed. In the first row, we can observe that some of these lifeforms hold very short flagella and are perfectly rounded shapes. In the second row, we can observe rounded shape lifeforms but with longer flagella. In the third row, we show another life form observed that holds a leaf shape with short flagella. The columns of intracellular forms feature selected images in which the background and surroundings of the parasite have been removed, enhancing the visibility of the parasite’s morphology. All these life forms are replicative and with short flagella, which aligns with the definition of amastigotes. In rows 2 and 3, the helical arrangement of microtubules from the cytostome cytoskeleton is observed (yellow arrows). ‘Bb’ stands for basal bodies; ‘CN’ stands for cell nucleus. All the images are maximum-intensity projections of selected z-stacks to allow for the visualization of the structures pointed. The scale bar is 1 µm.

**Figure 4 pathogens-13-00866-f004:**
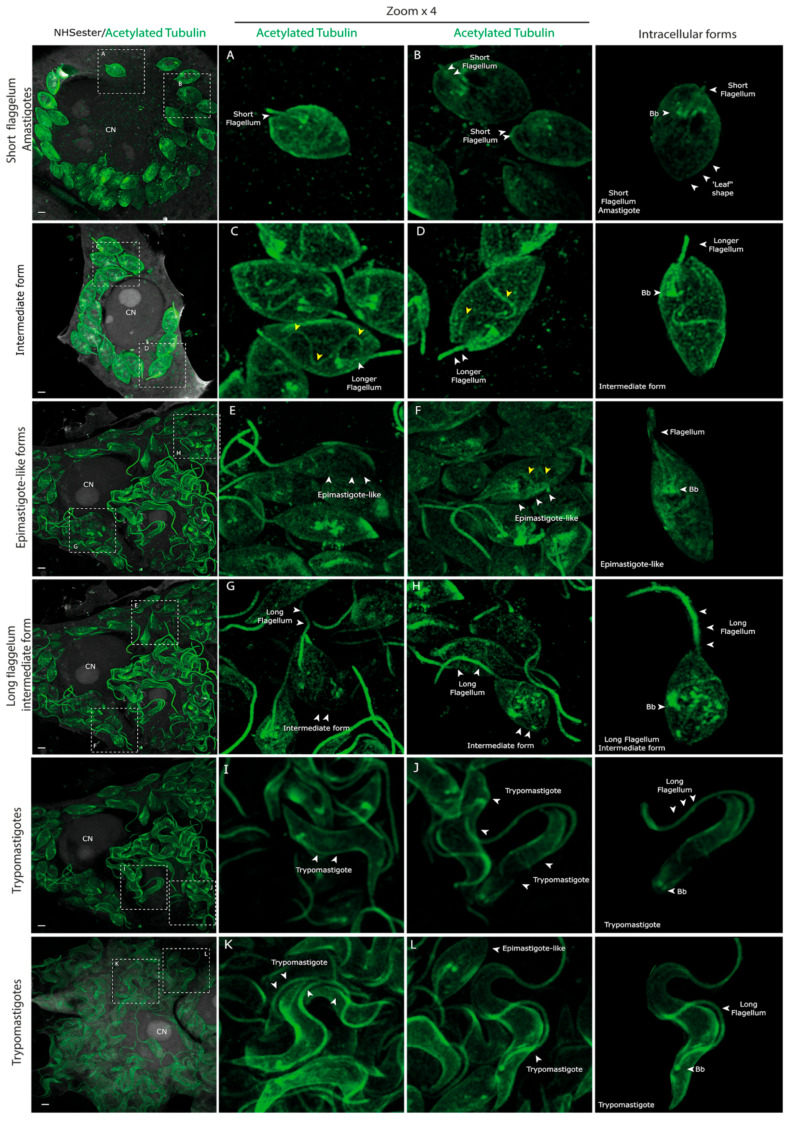
(**A**–**L**) Intracellular forms 72 h post-infection. At 72 h post-infection, not only amastigotes and trypomastigotes are observed. Consistent with previous observations, we observe epimastigote-like life forms, a ‘drop epimastigote’-like form and many intermediate amastigote forms, which meet the definition of amastigotes because of the position of the basal body, the kinetoplast, and nucleus. Still, they have a more sharpened shape instead of the rounded shape. All the images are maximum-intensity projections of selected z-stacks to allow for the visualization of the structures pointed. Please note that the columns of intracellular forms are selected images that have been cut using Inkscape V 1.3.2 (imaging software) to clarify their morphology observation. The helical arrangement of microtubules from the cytostome–cytopharynx cytoskeleton is indicated by yellow arrows. ‘Bb’ stands for basal bodies; ‘CN’ stands for cell nucleus. Please note that at 72 h, the number of parasites inside the cells was huge, and the DAPI signal tended to be faint and messy, making the visualization of the parasites difficult, which is why we decided not to include them in the figure. The scale bar is 1 µm in all images.

**Figure 5 pathogens-13-00866-f005:**
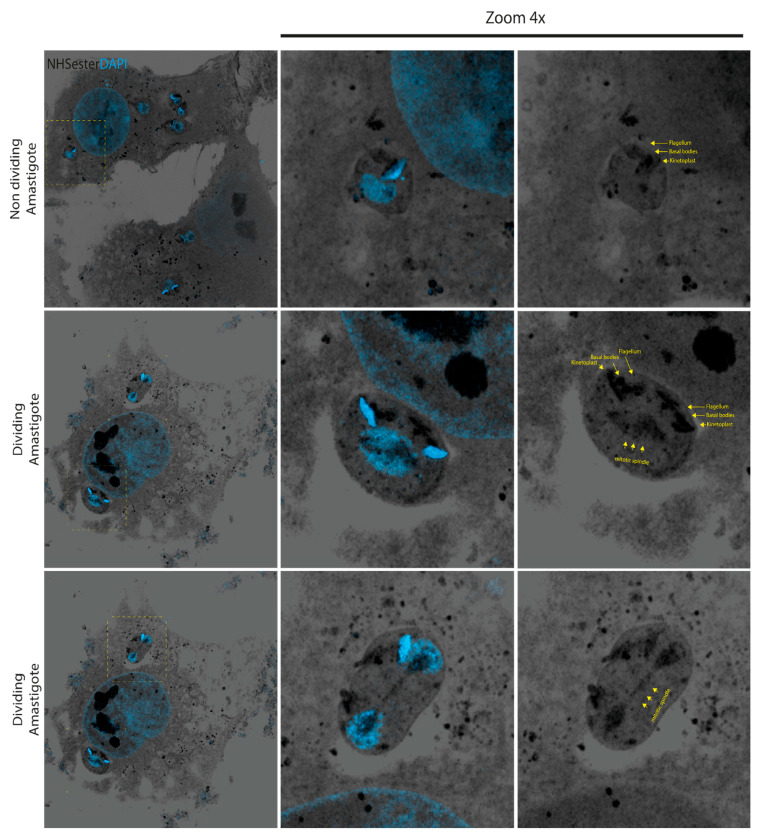
Amastigote cell division needs the assembly of a mitotic spindle nucleated without centrioles. During division, amastigotes assemble the mitotic spindle without disassembling the nuclear membrane and nucleating them without centrioles. Using UExM, we could observe in detail the binary fission process as never shown by immunofluorescence microscopy. Images were taken at 24 h post-infection. All the images are maximum-intensity projections of selected z-stacks to allow for the visualization of the structures indicated. The scale bar is 1 µm in all images.

**Figure 6 pathogens-13-00866-f006:**
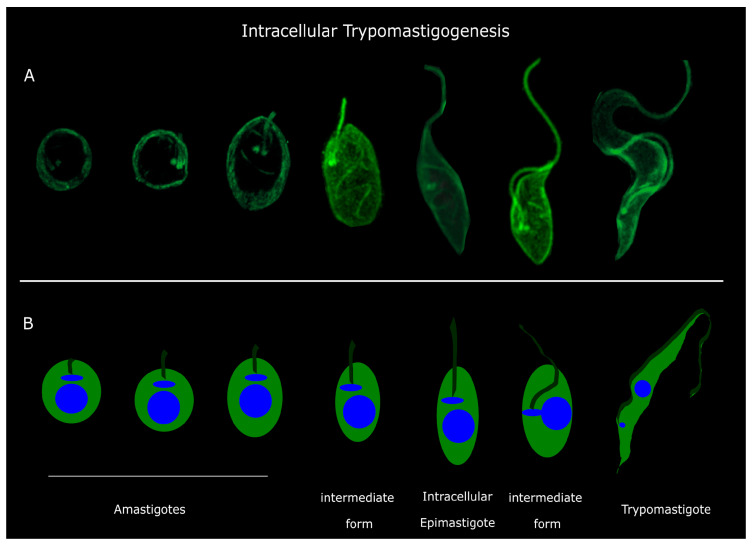
Intracellular trypomastigogenesis. (**A**) UExM of selecting different life forms observed 72 h post-infection. (**B**) Trypomastigogenesis model. As previously described, T. cruzi suffers several changes from amastigote to trypomastigote inside the infected cell. Using UExM, we could observe for the first time all these intermediate forms in detail that had never been obtained before using an immunofluorescence technique. The resolution achieved enabled us to identify two previously undescribed intermediate forms: one between the transition from amastigote to epimastigote and another between the transformation of epimastigotes into trypomastigotes, as illustrated in panels (**A**) and (**B**).

**Figure 7 pathogens-13-00866-f007:**
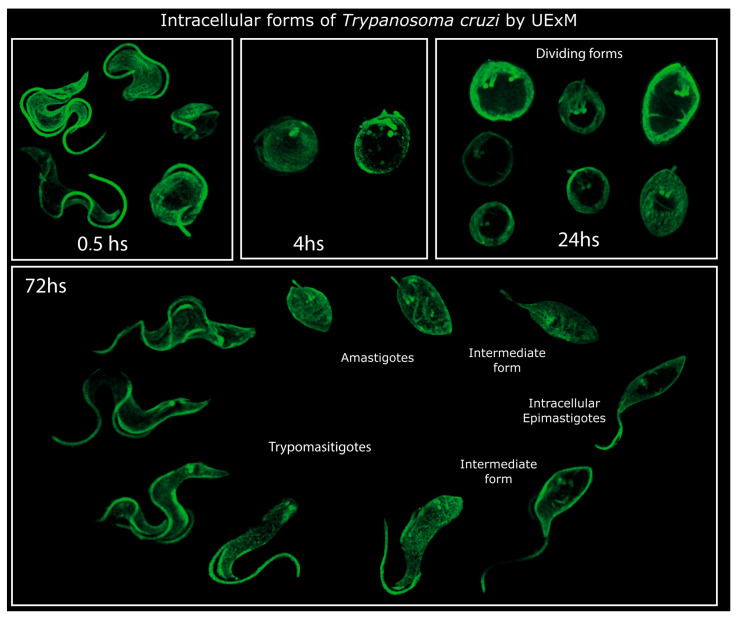
Intracellular life stages of Trypanosoma cruzi by UExM. UExM of selecting different life forms observed at 30 min, 4, 24, and 72 h post-infection. All the intracellular forms are selected images that have been cut using Inkscape V 1.3.2 (imaging software) to clarify the morphology observation.

## Data Availability

The original contributions presented in the study are included in article/Appendix A.

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
