# Peer review of "From Trypomastigotes to Trypomastigotes: Analyzing the One-Way Intracellular Journey of Trypanosoma cruzi by Ultrastructure Expansion Microscopy"

_pathogens, 2024, doi:10.3390/pathogens13100866_

Round 1

Reviewer 1 Report

Comments and Suggestions for Authors

Manuscript is clear in its objectives.

Very well written. Figures with good resolution, which facilitate the understanding of the work.

Bibliographic references 3, 11 and 27 are incomplete and need to be adapted.

Author Response

Very well written. Figures with good resolution, which facilitate the understanding of the work.

Bibliographic references 3, 11 and 27 are incomplete and need to be adapted.

We appreciate your review and comments. Now, the corrections are highlighted in the manuscript. In addition, we have substantially revised the English language (highlighted), references 3, 11, and 27, and Figures 2, 3,5, and 6 were modified.

Reviewer 2 Report

Comments and Suggestions for Authors

From Trypomastigotes to trypomastigotes: Analysing the one-way intracelular Journey of Trypanosoma cruzi by ultrastructure expansion microscopy

This article is interesting. It follows the journey of T. cruzi inside the host cell and accompanies the intracellular morphological transformations during the amastigogenesis and tripomastigogenesis process by Ultrastructure expansion microscopy. 

There are however, some points that were not clear to the reader and some points that would be better explored :

Materials and Methods :

In line 85 Trypomastigotes and amastigotes of T. cruzi (Dm 28c strain) were co-cultured in Vero cells...

·         The amastigotes and trypomastigotes forms were infected in VERO cells together in the same coverslips? What is the purpose of this procedure?

·         The article did not mention the proportion of parasites infected in relation to the cells.

    How long after the infection in VERO cells was the methodology for ultrastructure expansion microscopy performed ?

 Results:

·         Figure 1 showed trypomastigotes and rounded shapes parasite 30 minutes after infection. It appears that on observation of the images  the parasites are on top of the cell, not inside. The parasitophore vacuole was not observed and there are a lot of parasites infecting a single cell after 30 minutes. If the authors have performed co-cultivation  (tripomastigotes and amastigotes), the two forms will be found. If the parasites are on the cell and not inside, probably, the pH of culture medium will transform the tripomastigotes into amastigotes. The DAPI marking is too weak. 

·         In Figure 2, there was no DAPI marking. The authors should show more clearly  the kinetoplast position and nucleus shape if DAPI were used.

·         In figure 3, unfortunately, these images of the DAPI were also lacking. It would better for the authors to show the position and forms of the kinetoplast and nucleus. 

·     Figure 4 showed parasites intracellular forms after 72 hours of infection. The images are good and interesting and show diferent morphological forms of T. cruzi (amastigotes, epimastigotes-like, intermediate forms, tripomastogotes), in this case, the use of DAPI would be very interesting and enlightening because it could identify and differentiate the different stages of the parasites. 

·       Figure 5 The images which showed amastigotes in division are unclear . It could use the anti-tubulin antibody to facilitate identification.

·       Figure 6 As in the trypomastogogenesis model of the image in B, if the authors had used the DAPI to show the positions of the nucleus and kinetoplast it would be perfect.

·         Figure 7 Also missing  is the  use of DAPI because the images are very good.

Author Response

This article is interesting. It follows the journey of T. cruzi inside the host cell and accompanies the intracellular morphological transformations during the amastigogenesis and tripomastigogenesis process by Ultrastructure expansion microscopy. 

We appreciate your review and comments. The corrections are now highlighted in the manuscript. In addition, we have substantially revised the English language (highlighted), references 3, 11, and 27, and Figures 2, 3,5, and 6 were modified.

There are however, some points that were not clear to the reader and some points that would be better explored :

Materials and Methods :

In line 85 Trypomastigotes and amastigotes of T. cruzi (Dm 28c strain) were co-cultured in Vero cells…

We thank the reviewer for this observation, there was an error in the redaction. We have now modified all the paragraph (line 86)

The amastigotes and trypomastigotes forms were infected in VERO cells together in the same coverslips? What is the purpose of this procedure?

Vero Cells were infected with trypomastigotes. There was an error in the redaction. We have now modified all the paragraph (line 86)

The article did not mention the proportion of parasites infected in relation to the cells.

How long after the infection in VERO cells was the methodology for ultrastructure expansion microscopy performed ?

In the corrected paragraph, we explain the missing information.

 Results:

Figure 1 showed trypomastigotes and rounded shapes parasite 30 minutes after infection. It appears that on observation of the images  the parasites are on top of the cell, not inside. The parasitophore vacuole was not observed and there are a lot of parasites infecting a single cell after 30 minutes. If the authors have performed co-cultivation  (tripomastigotes and amastigotes), the two forms will be found. If the parasites are on the cell and not inside, probably, the pH of culture medium will transform the tripomastigotes into amastigotes. The DAPI marking is too weak. 

The parasites are located inside the cell, as demonstrated using Agave 3D, as previously shown in Fig. S1. We have also demonstrated that it is possible to distinguish whether the parasite is inside or outside the cell using 3D modeling and UExM images. In addition, we have modified Figure 2, and DAPI staining can now be observed, without the interference of the NHS ester signal.

In this figure (not in the manuscript) the intracellular and extracellular locations are clear. : Maximum intensity of z projection of cell infected with T. cruzi under UExM  B) This figure displays a three-dimensional model of a cell infected with parasites, reconstructed using Agave 3D. ‘i’ stands for intracellular parasite and ‘E’ stands for extracellular.

In Figure 2, there was no DAPI marking. The authors should show more clearly  the kinetoplast position and nucleus shape if DAPI were used.

We agree with this comment. We have now modified the Figure 2.

In Figure 3, unfortunately, these images of the DAPI were also lacking. It would better for the authors to show the position and forms of the kinetoplast and nucleus. 

We agree with this comment. We have modified Figure 3.

Figure 4 showed parasites intracellular forms after 72 hours of infection. The images are good and interesting and show diferent morphological forms of T. cruzi (amastigotes, epimastigotes-like, intermediate forms, tripomastogotes), in this case, the use of DAPI would be very interesting and enlightening because it could identify and differentiate the different stages of the parasites. 

Although we agree, there is a technical issue: at 72 h the large number of parasites inside the cells significantly reduces the DAPI signal and tends to be faint and unclear. Therefore, we performed this experiment without DAPI, using NHS ester, which provides clear cell morphology. Observing the position of the basal bodies and the cell body, along with the information presented in Figs. 1, 2, and 3, the kinetoplast and nucleus’s morphology and location can be determined.

Figure 5 The images which showed amastigotes in the division are unclear. It could use the anti-tubulin antibody to facilitate identification.

We understand the point. However, Figure 5 complements Figure 3, which shows similar data with Tubulin. We propose keeping Figure 5.

  •      Figure 6 As in the trypomastogogenesis model of the image in B, if the authors had used the DAPI to show the positions of the nucleus and kinetoplast it would be perfect.

As explained above, at 72 h, the large number of parasites inside the cells significantly reduces the DAPI signal, which tends to be faint and unclear.

Figure 7 Also missing  is the  use of DAPI because the images are very good.

As shown in Fig. 7, we have several parasites from different figures (1, 3, 4). Since Fig. 4 does not include the DAPI channel, as previously explained in the comments, we decided to maintain the presentation as originally shown. We understand that this approach allows for better observation of the parasite morphology.

Reviewer 3 Report

Comments and Suggestions for Authors

This work aims to elucidate life cycle transitions of Trypanosoma cruzi in infected mammalian cells. Authors employ Ultrastructure expansion microscopy, an approach which facilitates this type of studies, as it enables identification of infrequent cell types, such as various transient life cycle stages, and their imaging at a significantly increased resolution compared to classic immunofluorescence, therefore providing substantially more information on cell morphology and organization. The authors used the approach to study differentiation of trypomastigotes to amastigotes and amastigotes to trypomastigotes via epimastigotes. These are important yet poorly understood aspects of biology of T. cruzi, which are critical for the disease it causes. Hence, the manuscript is potentially very interesting and aims to provide original findings.

However, I have major concerns, particularly regarding the imaging data and their interpretation. Many images in the manuscript do not appear to represent entire cellular volume- for example Figure 1E and F, Figure 3, Figure 4B, E, and H etc. This may be partially caused by the authors representing Z-stacks like maximum projections, which may explain differences in the intensity of the corset microtubules. Or maybe the authors on purpose removed top and bottom parts of the cells to be able to visualize cellular interior? This is unclear and the authors should explain this, and show entire cells in addition to partial volumes, ideally as 3D reconstructions. This is really problematic when it comes to the flagellum, or better its microtubule cytoskeleton, which the authors visualize (and should acknowledge this). In Fig 1B, E, and F parts of flagella are clearly missing (the flagellar signal is discontinuous). The cell in Figure 1E appears to have two parallel flagella while the one in Fig. 1F has a long and short flagellum... Importantly, because of this, it is difficult to tell where exactly flagella terminate, and therefore how long they are. Yet, this is one of criteria used to determine a life cycle stage of cells. In addition, the authors should measure lengths of flagella in individual cells, rather than just categorizing them into short and long flagella. Moreover, based on the data presented, I am not convinced that a cell folds around itself, allowing contact between its posterior and anterior parts, as mentioned several times, e.g. line 121 – for this the authors would really have to show a 3D reconstruction of several such cells from different angles and also an transient state with fusing ends.

Similarly, position of the kinetoplast and nucleus are hallmarks of individual life cycle stages, yet the authors do not show the corresponding DNA stain signal in most of the images. Hence, positions of the organelles are unknown.

I did not find any estimate of the expansion factor. It is not clear whether scale bars in images refer to a physical size of expanded specimens.

Many statements are not precise or should be further explained, for example:

85 - Trypomastigotes and amastigotes of T. cruzi (Dm28c strain) were co-cultured in Vero cells, - could the authors explain how this was done, which culture was used for which experiment?

104 – NHS ester 594 (Invitrogen) ... used to label the amino acid bond – does the NHS ester indeed label the amino acid bond?

108 – could the authors specify the camera used for imaging?

Figure 1 – some of the images (A-E, B-D) are identical, and it is not clear, why the figure is arranged like this and not like Figure 4 (except the 3rd and 4th image from top (left) in Fig. 4., which are again identical?), where multiple cells are enlarged from a single image.

190 - The column of intracellular forms is made up of selected images that have been cut using imaging software -  could the authors rephrase this?

153 - All these life forms are observed in the process of the observation of amastigotes – can you explain this?

211- what are ‘drop epimastigote’ like form?

226 - Almost reaching the last step before mature trypomastigotes are visualized, the presence of epimastigote-like forms appears (Fig. 4). – rephrase

Figure 4- it is flagellum, not flaggellum

Figure 5 – middle and bottom row – wrong areas of magnification indicated

267 - binary fission – I am used to use this term when referring to bacteria, check whether is it a correct term

315 – these cells are called zoids not zoites

316 - In our work, we did not observe either the biflagellated cell or the zoite. – can you hypothesize why you did not observe them?

332 – 339 This paragraph only recapitulates what was said previously.

Figure 6A, the rightmost cell – does it have two flagella?

Figure 7, 24 hours – based on which criteria are some of these cells (e.g. bottom left) dividing?

Some of the references (3, 21) are incomplete. All should be carefully checked.

Supplementary-

while the second panel on the right corner – please rephrase

Figure S2. 4hs post infection amastigotes inside the host cell are observed. Using UExM we could observe in detail the intracellular life form present at 4hs post infection. At 4 hours post-infection, amastigotes are observed. – what does this mean?

Comments on the Quality of English Language

Overall, English is good, but there are cases when words are misspelled, parts of text are difficult to understand etc. - see Suggestions for Authors above. Also more attention should be paid when figures assembled, to references etc.

Author Response

This work aims to elucidate life cycle transitions of Trypanosoma cruzi in infected mammalian cells. Authors employ Ultrastructure expansion microscopy, an approach which facilitates this type of studies, as it enables identification of infrequent cell types, such as various transient life cycle stages, and their imaging at a significantly increased resolution compared to classic immunofluorescence, therefore providing substantially more information on cell morphology and organization. The authors used the approach to study differentiation of trypomastigotes to amastigotes and amastigotes to trypomastigotes via epimastigotes. These are important yet poorly understood aspects of biology of T. cruzi, which are critical for the disease it causes. Hence, the manuscript is potentially very interesting and aims to provide original findings.

We appreciate your review and comments. The corrections are now highlighted in the manuscript. In addition, we have substantially revised the English language (highlighted), references 3, 11, and 27, and Figures 2, 3,5, and 6 were modified. 

However, I have major concerns, particularly regarding the imaging data and their interpretation. Many images in the manuscript do not appear to represent entire cellular volume- for example Figure 1E and F, Figure 3, Figure 4B, E, and H etc. This may be partially caused by the authors representing Z-stacks like maximum projections, which may explain differences in the intensity of the corset microtubules. Or maybe the authors on purpose removed top and bottom parts of the cells to be able to visualize cellular interior? This is unclear and the authors should explain this, and show entire cells in addition to partial volumes, ideally as 3D reconstructions. This is really problematic when it comes to the flagellum, or better its microtubule cytoskeleton, which the authors visualize (and should acknowledge this). In Fig 1B, E, and F parts of flagella are clearly missing (the flagellar signal is discontinuous). The cell in Figure 1E appears to have two parallel flagella while the one in Fig. 1F has a long and short flagellum... Importantly, because of this, it is difficult to tell where exactly flagella terminate, and therefore how long they are. Yet, this is one of criteria used to determine a life cycle stage of cells. In addition, the authors should measure lengths of flagella in individual cells, rather than just categorizing them into short and long flagella. Moreover, based on the data presented, I am not convinced that a cell folds around itself, allowing contact between its posterior and anterior parts, as mentioned several times, e.g. line 121 – for this the authors would really have to show a 3D reconstruction of several such cells from different angles and also an transient state with fusing ends.

We thank the reviewer for these comments and suggestions. Maximum Z-projection of selected Z slices was used better to visualize certain structures, such as basal bodies. The length of the flagella is compared with the cell body itself. The contact between the posterior and anterior parts is also observed in Fig. 1C, 1D, and 1E. This has been demonstrated using higher resolution techniques, such as TEM (Pan 1978, cited in this article). An article by Andrews et al. (1987) proposed a similar mechanism based on evidence collected using scanning electron microscopy, as cited in this article. We have added a small figure with 3D models of UExM images collected during the experiments. Of course, our ‘rolling motion’ model is just that—a model. We believe this process occurs rapidly, making documentation quite challenging, and probably should be part of another work through in vivo microscopy.

3D Model of Selected Parasites Generated Using Agave 3D Software. (Not included in the article) The figure illustrates a three-dimensional reconstruction of many parasites, with the posterior part (Pp) distinctly pointed. The visualisation was generated using the ac-tubulin channel. 

Similarly, position of the kinetoplast and nucleus are hallmarks of individual life cycle stages, yet the authors do not show the corresponding DNA stain signal in most of the images. Hence, positions of the organelles are unknown.

The position of the kinetoplast towards the nucleus is paramount in the definition of life stages; we corrected Figures 2 and 3 as suggested. At 72 h the number of parasites inside the cells is huge, and the  DAPI signal tends to be faint and messy. We decided to perform this experiment without DAPI but with NHS ester which gives you the cell morphology. By the position of the basal bodies and by the cell body and with the info shown in Fig1, 2, 3, you can tell the position of the kinetoplast and nucleus making easier the interpretation of the morphology of these life stages.

I did not find any estimate of the expansion factor. It is not clear whether scale bars in images refer to a physical size of expanded specimens. 
It was missing. We added this information to the text ( the expansion factor was the same as previously reported ~3,5-4). The scales bar are for the physical size of expanded specimens.

Many statements are not precise or should be further explained, for example:
85 - Trypomastigotes and amastigotes of T. cruzi (Dm28c strain) were co-cultured in Vero cells, - could the authors explain how this was done, which culture was used for which experiment?

Vero Cells were infected with trypomastigotes. There was an error in the redaction. We have now modified the paragraph (line 86).

104 – NHS ester 594 (Invitrogen) ... used to label the amino acid bond – does the NHS ester indeed label the amino acid bond?
It was wrong.
Corrected …“label primary amines on proteins.”

108 – could the authors specify the camera used for imaging?

The microscope used for the experiments was Zeiss LSM800 confocal microscope (Oberkochem, Germany) with a Plan-Apochromat 63×/1.40 oil immersion objective.

Figure 1 – some of the images (A-E, B-D) are identical, and it is not clear, why the figure is arranged like this and not like Figure 4 (except the 3rd and 4th image from top (left) in Fig. 4., which are again identical?), where multiple cells are enlarged from a single image.

We consider it makes easier the interpretation of what we want to show. 

190 - The column of intracellular forms is made up of selected images that have been cut using imaging software -  could the authors rephrase this?

The columns of intracellular forms feature selected images in which the parasite’s background and surroundings have been removed, enhancing the visibility of its structure. We have now corrected this.

153 - All these life forms are observed in the process of the observation of amastigotes – can you explain this?

In the passage to amastigotes, we propose in the text that T. cruzi needs to go through all these intermediate stages.

211- what are ‘drop epimastigote’ like form?

It is a life form observed in epimastigogenesis in the vector with a particular drop like shape that was previously reported ( references 4 and 29)

226 - Almost reaching the last step before mature trypomastigotes are visualized, the presence of epimastigote-like forms appears (Fig. 4). – rephrase

Just before the emergence of mature trypomastigotes, epimastigote-like forms are observed. We changed it in the Ms.

Figure 4- it is flagellum, not flagellum

We have now corrected it in Figure 4. 

Figure 5 – middle and bottom row – wrong areas of magnification indicated

We agree with this observation. It was wrong. We have now modified them in Figure 5.

267 - binary fission – I am used to use this term when referring to bacteria, check whether is it a correct term
We agree that binary fission is usually used wher referring to bacteria. However, the term is also used in these parasites. As an example, here are some references:

https://www.ncbi.nlm.nih.gov/pmc/articles/PMC7112235/
https://www.cdc.gov/dpdx/trypanosomiasisamerican/index.html
https://www.ncbi.nlm.nih.gov/pmc/articles/PMC3556388/
American Trypanosomiasis (Chagas disease) Rogelio López-Vélez, ... Caryn Bern, in Hunter's Tropical Medicine and Emerging Infectious Disease (Ninth Edition), 2013

315 – these cells are called zoids not zoites

Yes. We have now corrected this mistake. 

316 - In our work, we did not observe either the biflagellated cell or the zoite. – can you hypothesize why you did not observe them?

In our opinion, those biflagellated/zoite images were an artefact. No one ever described it again, and all the citations of that paper refer to the immune response and not to the supposed biflagellate. 
In short, for us, it does not exist mainly because no one else documented them.

We have now modified the text and added this comment.

332 – 339 This paragraph only recapitulates what was said previously.

Yes, we agree. But we propose to keep it to summarize the article just before the last paragraph.

Figure 6A, the rightmost cell – does it have two flagella?

No, it is a trypomastigote.Based on your comment, we have now modified Figure 6, selecting another trypomastigote to make this clearer. 

Figure 7, 24 hours – based on which criteria are some of these cells (e.g. bottom left) dividing?

Based on the number of basal bodies, which can be observed using UEXM, you can tell whether they are dividing forms or not. If you can observe 4 basal bodies, these are dividing forms. If you always observe only 2 basal bodies, they at least seem not to be replicative forms..

Some of the references (3, 21) are incomplete. All should be carefully checked.

We have now corrected them.

Supplementary-
while the second panel on the right corner – please rephrase
Figure S2. 4hs post infection amastigotes inside the host cell are observed. Using UExM we could observe in detail the intracellular life form present at 4hs post infection. At 4 hours post-infection, amastigotes are observed. – what does this mean?

It means that at 4 hours post-infection, we only observed amastigotes but not trypomastigotes inside the cell.

Comments on the Quality of English Language
Overall, English is good, but there are cases when words are misspelled, parts of text are difficult to understand etc. - see Suggestions for Authors above. Also more attention should be paid when figures assembled, to references etc.

We have substantially revised the English language (highlighted), references 3, 11, and 27, and Figures 2, 3,5, and 6 were modified. 

Round 2

Reviewer 1 Report

Comments and Suggestions for Authors

Well-structured manuscript that reveals interesting data about the relationship of T. cruzi forms with the host cell.

Author Response

We acknowledge the reviewer's new revision and her/his consideration that this is a well-structured manuscript with interesting information about the relationship of T. cruzi forms with the host cell.

Reviewer 3 Report

Comments and Suggestions for Authors

I appreciate that the authors considered my suggestions and made changes when they thought it was appropriate.

However, my major concern still to some extent persists- I appreciate that the authors are presenting ‘Maximum Z-projection of selected Z slices was used better to visualize certain structures, such as basal bodies.’ – this is an appropriate thing to do to show signal within the cell (within the cytoskeleton). However, this is not appropriate to demonstrate the cell shape, the position of the flagellum on the cell surface, and its length. Yet, these are features critical for the reader to understand what is happening. Ideally, both should be shown (at least for cells where it is clear that a part of volume is missing) – 3D reconstructions of cells to demonstrate their shape, the path of the flagellum on the surface, and the flagellar length, and projection of selected slices to highlight structures inside. For example, I still do not understand how cells magnified in 1E and 1F look like- e.g. do they have a single flagellum or two flagella? The cell in 1E has the basal body indicated, but there is no DAPI signal of an associated kinetoplast?

 ‘The length of the flagella is compared with the cell body itself.’ – I still think the correct thing is to measure flagellar lengths and classify cells in an unbiased manner.

Unfortunately, I was not able to look at the mentioned 3D Model of Selected Parasites Generated Using Agave 3D Software via the journal interface.

I appreciate that the authors included DAPI signals in Figs 2 and 3. It may be a good idea to mention in the text why it was not possible to include the signal also in the subsequent figures.

108 – could the authors specify the camera used for imaging?

‘The microscope used for the experiments was Zeiss LSM800 confocal microscope (Oberkochem, Germany) with a Plan-Apochromat 63×/1.40 oil immersion objective.’  – I still do not know which type of camera was used. The type should be stated also in the manuscript.

Figure 7, 24 hours – based on which criteria are some of these cells (e.g. bottom left) dividing?

‘Based on the number of basal bodies, which can be observed using UEXM, you can tell whether they are dividing forms or not. If you can observe 4 basal bodies, these are dividing forms. If you always observe only 2 basal bodies, they at least seem not to be replicative forms..’ -  I agree with this explanation. However, I cannot see 4 basal bodies in some of the cells, for example bottom left and bottom centre.

Round 3

Reviewer 3 Report

Comments and Suggestions for Authors

Some issues have been resolved. The following persist:

1/ 3D Model of Fig1F  Generated Using Agave 3D Software. – I assume a 3D volume of one cell has been reconstructed – and is presented in 4 different views/angles. The cell may indeed have a single flagellum. However, for some reason the axonemal signal is discontinuous; it loops around the cell plus there is a part not attached to the cell, but no signal in between. Clearly, a part of the cell volume is missing in the imaging. This poses problems to interpretation. Here I copy a part of my review for the first round of revisions, which still applies:

‘In Fig 1B, E, and F parts of flagella are clearly missing (the flagellar signal is discontinuous). Figure 1E appears to have two parallel flagella while Fig. 1F has a long and short flagellum? Importantly, because of this, it is difficult to tell where exactly flagella terminate, and therefore how long they are. Yet, this is one of criteria used to determine a life cycle stage of cells.‘

Hence, I would not be able to unambiguously interpret these data, particularly when it comes to the flagellum length. I do not see a reason why not showing 3D reconstructions of entire volumes of the cells in question, for the reader to be able to form his/her opinion on the path of the flagellum and its length. Such 3D reconstructions can be easily performed using a free software, such as ImageJ (the 3D viewer plugin). However, for a complete 3D reconstruction an imaging of an entire cell volume is needed. Maybe there are technical reasons why this is difficult to achieve in these samples- is that so? Could the authors select cells, for which complete cell volumes are available?

2/ I did not find any comment on the following: ‘The length of the flagella is compared with the cell body itself.’ – I still think the correct thing is to measure flagellar lengths and classify cells unbiasedly.‘ – I do realize that measuring distances in 3D images is not trivial.

3/ My last point - Figure 7, 24 hours – based on which criteria are some of these cells (e.g. bottom left) dividing?

‘Based on the number of basal bodies, which can be observed using UEXM, you can tell whether they are dividing forms or not. If you can observe 4 basal bodies, these are dividing forms. If you always observe only 2 basal bodies, they at least seem not to be replicative forms..’ -  I agree with this explanation. However, I cannot see 4 basal bodies in some of the cells, for example bottom left and bottom centre.‘

The authors claim that: ‘The bottom left panel includes trypomastigotes, which are non-dividing forms. In the center, the forms displayed are amastigotes, which are dividing forms, as demonstrated in the bottom right and bottom center panels. Therefore, in our opinion, the absence of four basal bodies is consistent with these observations.‘

Yet, I was referring to the Figure 7, 24 hours image, as stated above. This figure does not contain any trypomastigotes, and is labeled ‘Dividing forms‘. Yet, I cannot see 4 basal bodies in some of the cells, for example bottom left and bottom centre (of Figure 7, 24 hours). Could the authors label the 4 basal bodies in these cells to help me to see them?
